# In-context Learning with Retrieved Demonstrations for Language Models: A Survey

**Man Luo**    man.luo@intel.com
*Intel Lab.*

**Xin Xu**    xxujasmine@google.com
*Google Research*

**Yue Liu**    yliujune@google.com
*Google Research*

**Panupong Pasupat**    ppasupat@google.com
*Google Research*

**Mehran Kazemi**    mehrankazemi@google.com
*Google Research*

**Reviewed on OpenReview:** *https://openreview.net/forum?id=NQPo8ZhQPa*

## Abstract

Large language models have demonstrated remarkable few-shot in-context learning (ICL) capabilities, adapting to new tasks with few-shots demonstrations. However, the efficacy of ICL is highly dependent on the selection of these demonstrations. Recent developments have introduced retrieval-based in-context learning (RetICL), which dynamically retrieves demonstrations tailored to each input query. This approach leverages existing databases and retrieval systems, enhancing efficiency and scalability while mitigating biases inherent in manual example selection. Given the promising results and growing interest in RetICL, we present a comprehensive survey of this field. Our review encompasses: design choices for ICL demonstration retrieval models, retrieval training procedures, inference strategies and current applications of RetICL. In the end, we explore future directions for this emerging technology.

## 1 Introduction

Few-shot in-context learning (ICL) is the ability of large language models (LLMs) to perform a new task when a few input-output examples, or *demonstrations*, for the new task are given alongside the actual task input. Importantly, the model parameters do not have to be fine-tuned towards the new task. ICL is popularized by the work on pre-trained large language models, which can perform ICL without being trained to do so (Brown et al., 2020), though smaller language models can also be explicitly trained to perform ICL (Min et al., 2022a).

ICL presents several advantages over the conventional methodology for adapting language models to a downstream task, which typically involves initial pre-training followed by subsequent fine-tuning. One significant merit of ICL is the circumvention of fine-tuning, which might not always be possible due to limited access to the model parameters or constraints on computational resources (Brown et al., 2020). Furthermore, ICL avoids common issues associated with fine-tuning, such as overfitting (Ying, 2019; Kazemi et al., 2023a). Compared to parameter-efficient fine-tuning methods (PEFT) (Hu et al., 2021; Dettmers et al., 2023; Lester et al., 2021), ICL is computationally cheaper and remain the model parameters unchanged thus preserving the generality of the LLMs.

Early ICL implementations use a fixed set of demonstrations for each target task. These demonstrations could be hand-crafted by human (Hendrycks et al., 2021; Wei et al., 2022; Kazemi et al., 2023b) or randomly chosen from training data (Brown et al., 2020; Lewkowycz et al., 2022). Beyond random selection, there are more advanced selection processes based on metrics such as complexity (Fu et al., 2022), diversity (Li & Qiu, 2023a), difficulty (Drozdov et al., 2023), concept learning (Wang et al., 2023b) and perplexity (Gonen et al., 2023). Importantly, the demonstrations remain *context-insensitive* (i.e. the same demonstrations are used regardless of the query) which could hinder unlocking the true potential of the LLMs. The effectiveness of such demonstrations is influenced by factors such as the quality, quantity, and ordering of the demonstrations (Brown et al., 2020; Naik et al., 2023; Kojima et al.; Lu et al., 2022b).

Retrieval-based ICL (RetICL) presents a paradigm shift in the optimization of language model performance, moving beyond static, pre-defined demonstration sets to a dynamic, context-sensitive approach. At the heart of this innovation is the concept of *adaptive demonstration selection*, where a specialized retriever intelligently curates tailored demonstrations for each specific task input. This method has not only consistently outshined approaches relying on random or static hand-crafted demonstrations (Liu et al., 2022; Rubin et al., 2022; Luo et al., 2023) but has also demonstrated a remarkable resilience to a variety of influencing factors (Li et al., 2023b). RetICL shares similarities with the broader concept of Retrieval Augmented Generation (RAG) (Guu et al., 2020; Lewis et al., 2020; Izacard et al., 2023). Both approaches aim to retrieve external information to augment the prompt and improve model inference performance. However, RAG encompasses a wider spectrum of retrieval methods and use cases. While RetICL specifically focuses on retrieving demonstrations for in-context learning, RAG can involve retrieving various types of relevant information in response to a query, not necessarily limited to demonstrations.

The efficacy of RetICL pivots on the "relevance" and "usefulness" of the demonstrations it selects, a process intricately influenced by multiple elements. These include the nature of the retriever—ranging from general off-the-shelf models to finely-tuned, domain-specific variants—the source and diversity of the retrieval corpus, the retriever's objectives (focusing on either similarity or diversity), and the strategies for integrating multiple demonstrations. Over the past two years, numerous and sometimes concurrent works have studied RetICL each with different terminology and with variations in problem definition and subsequent methodologies, making it difficult to comprehend the current state of research and practice in RetICL, especially for newcomers to the field. In this comprehensive survey, we meticulously analyze 22 seminal papers in the field of RetICL, as detailed in Table 1[1], and provide a categorization of their main building blocks (See Figure 1). Our work not only provides a thorough synthesis of existing research but also underscores the areas where RetICL significantly surpasses previous ICL methods, and illuminates many paths forward for future innovations in this area, thus serving as a critical resource for ICL.

## 2 Few-shot In-context Learning for Language Models

Language models (LMs) (Zhao et al., 2023; Rosenfeld, 2000; Jurafsky, 2021; Radford et al., 2019; Raffel et al., 2020; Lewis et al., 2019; Touvron et al., 2023) are probabilistic models that assign probabilities to sequences of words and are essential components in many tasks. Let $s$ represent a sequence of words (e.g., a sentence) and $w_1, w_2, \ldots, w_n$ represent the tokens in the sequence. Based on the chain rule, the probability $p(s)$ can be decomposed into the following product of probabilities:

$$p(s) = p(w_1)p(w_2 \mid w_1) \ldots p(w_n \mid w_1, \ldots, w_{n-1}) = \prod_{k=1}^{n} p(w_k \mid w_1, \ldots, w_{k-1})$$

where each element in the product corresponds to the probability of a token given the previous tokens. Based on the above decomposition, an LM can be constructed by learning the probability of the next token given the previous ones.

Earlier LMs were mostly based on N-gram models (Jurafsky, 2021) and has been used in computing approximations to English word sequences (Shannon, 1948) and speech recognition system (Baker, 1990; Jelinek et al., 1975; Baker, 1975; Bahl et al., 1983; Jelinek, 1990). N-gram models are based on the

---

[1]The list of papers in this table can be found in here `https://github.com/luomancs/luomancs-reticl_llm_survey/`.

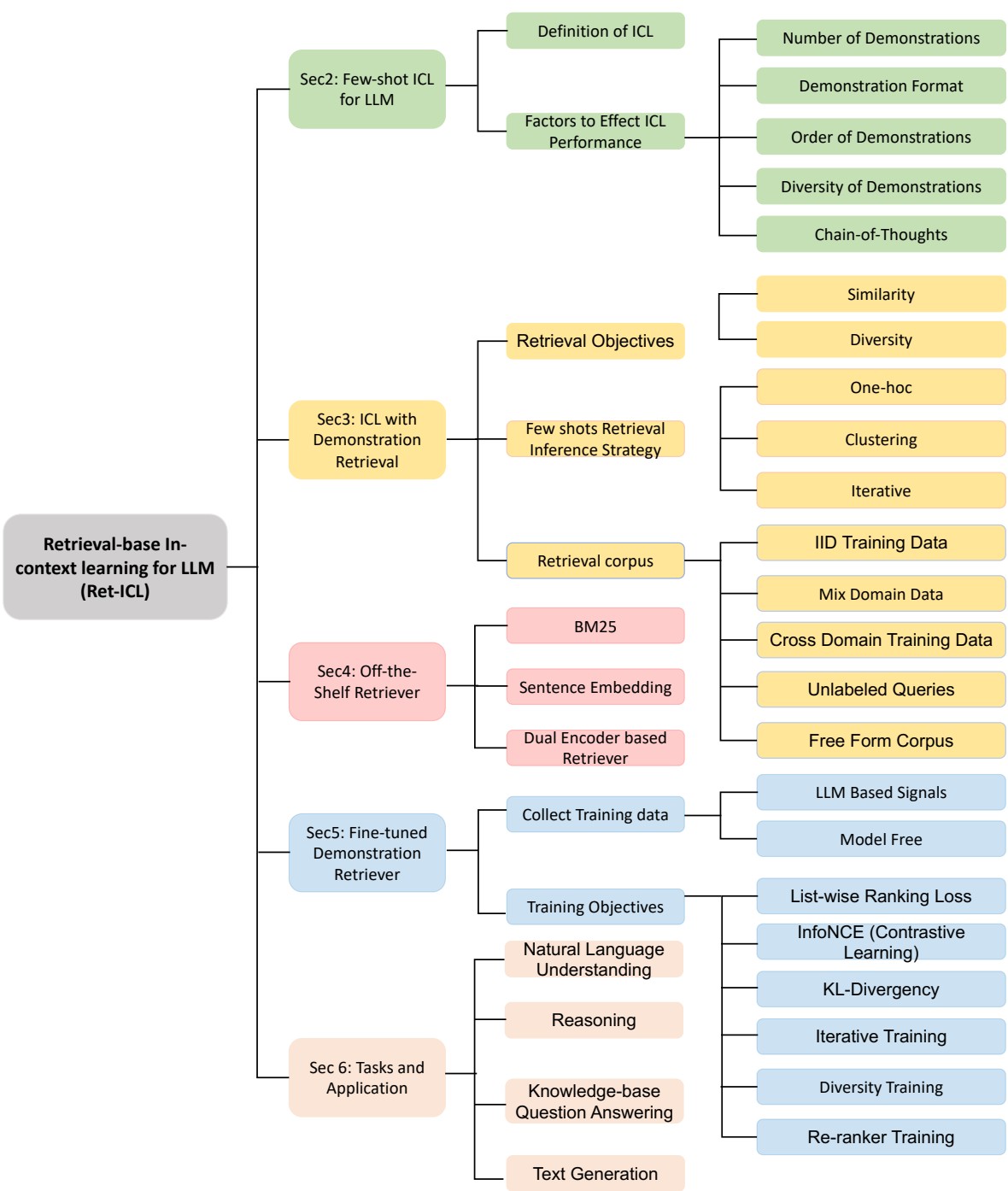

Figure 1: Structure of the Survey.

| Paper | LLMs | Retrieval Method | Retrieval Corpus | Evaluation Tasks | Retriever Training | Retrieval Strategy |
|---|---|---|---|---|---|---|
| SYNCHRO MESH 2021 | GPT-3 | SBERT + Target Similarity | In Domain | CodeGen | Target Similarity Tuning | Top-k |
| KATE 2022 | GPT-3 | RoBERTa+kNN | In Domain | SA, Table2Text, QA | Sentence Similarity Embedding | Top-k |
| EPR 2022 | GPT-J, CODEX, GTP-3 | SBERT, BM25, Fine-tuned Retriever | In Domain | SP | InfoNCE Loss(contrastive learning) | Top-k |
| Z-ICL 2022 | GPT-J, GPT-NeoX, GPT-3 | SimCSE (Sentence Embeddings) | Free Form Corpus | SA | Sentence Embeddings Similarity | Top-k |
| Mem Prompt 2022 | GPT-3 | Query Transformation + SBERT | In Domain with Human Feedback | Word Reasoning, Ethical Reasoning | Sentence Similarity Embeddings | Top-k |
| Teach Me 2022 | T5-11B | BM25 | In Domain with Human Feedback | QA | Term-based Similarity | Top-k |
| IC-DST 2022 | GPT-3 | Fine-tuned SBERT Retriever | In Domain | DST-to-SQL | Contrastive Learning | Top-k |
| Vote-k 2022 | GPT-Neo, GPT-J,GPT-3, CODEX | SBERT | Selected subsets of In Domain, labeled | SA, NLI, QA, Summ, CSR, RC | Sentence Embeddings Similarity | Top-k |
| Auto-CoT 2022b | GPT-3 | Clustering with SBERT | LLM Generated CoT for In Domain | MathR, QA | Sentence Embeddings Similarity | Clustering |
| XRICL 2022 | CODEX | Fine-tuned Retriever Combined with a Re-ranker | Cross Domain | Text-to-SQL | Distillation by KL Divergence | Top-k |
| DSP 2022 | GPT-3 | ColBERTv2 | In Domain | QA | Token Embeddings Similarity | Iterative |
| PARC 2022 | mBERT | Multilingual SBERT | Cross Domain | SA, TopicC, NLI | Sentence Embeddings Similarity | Top-k |
| Cover-LS 2022 | CODEX, T5-large | BM25 and Diversity Selection | In Domain | SP | Diversity Training | Iterative |
| Prompt PG 2022a | GPT-3 | Fine-tuned Retriever | In Domain | MathR | Policy Gradient | Top-k |
| Dynamic Least-to-Most 2022 | DODEX | Tree Structure Similarity | In Domain | SP | Diversity and Lexical Overalp | Iterative |
| Self-Prompting 2022a | Instruct GPT, CODEX | SBERT with Clustering | Free Form Corpus | QA | Sentence Embeddings Similarity | Clustering |
| CEIL 2023a | GPT-NEO, GPT2-XL, CODEX | BM25, BERT, Fine-tuned Retriever | In Domain | SA, PD, NLI, CSR, QA, CodeGen, and SP | Diversity Training | Iterative |
| R-BM25 2023 | XGLM-7.5G | BM25, BM25 and Term Recall based rerank In Domain | In Domain | MT | Term-based Similarity | Top-k |
| ICL-MC 2023 | GPTNeo-2.7B, XGLM-2.9G, BLOOM-3B | BM25 | In Domain Cross Domain | MT | Term-based Similarity | Top-k |
| ICL-ML 2023 | OPT-13/175B, LLaMA-7/70B | SBERT | In Domain | Intent Classification | Sentence Embeddings Similarity | Top-k |
| UP RISE 2023 | GPT-Neo, BLOOM, OPT, GPT-3 | Fine-tuned Retriever | Cross Domains Human Feedback | RC, QA, NLI, SA, CSR, CR, PD | Contrastive Learning | Top-k |

| Paper | LLMs | Retrieval Method | Retrieval Corpus | Evaluation Tasks | Retriever Training | Retrieval Strategy |
|---|---|---|---|---|---|---|
| Dr.ICL 2023 | PaLM, Flan-PaLM | BM25, GTR, Fine-tuned Retriever | In Domain | QA, NLI, MathR | Contrastive Learning | Top-k |
| LLM-R 2023a | LLaMA-7B | Fine-tuned Retriever and Reward Model | Mix Domain | QA, CSR, CR, PD, RC, SA, D2T, Summ, NLI | Contrastive learning + KL | Top-k |
| UDR 2023b | GPT-J, GPT-Neo, CODEX, GPT-3 | Fine-tuned Unified Retriever with Iterative Data Mining | Mix Domain | SA, TC, CSR, NLI, SP, StoryGen, Summ, D2T | Contrastive Learning | Top-k |
| MoT 2023b | ChatGPT | SBERT with Clustering and Filtering, LLM as Retriever | Unlabelled Queries with LLMs CoT | MathR, NLI, CSR, QA | Sentence Embeddings Similarity | Clustering |
| RetICL 2023 | CODEX | Iterative LSTM with Reinforcement Learning | In Domain | MathR | PPO and GAE | Iterative |
| Ambig-ICL 2023 | Flan-PaLM | Fine-tuned multilingual T5, and a Filtering Algorithm Based on LLM feedback | In Domain | TC, SA | Contrastive Learning | Top-k |

Table 1: Comparison with Related Work. Abbreviation for Evaluation Tasks: CodeGen (code generation), SA (sentiment analysis), Table2Text (Table to Text generation), QA (question answering), SP (semantic parsing), DST (Dialogue State Tracking), D2T (Data-to-Text), Summ ( Summarization), CSR ( commonsense reasoning), RC (reading comprehension), NLI (natural language inference), CR (Coreference Resolution), MathR (mathematical reasoning), PD (paraphrase detection), TQA (Table Question Answering), TC (Topic Classification), StoryGen (Story Generation), MT (Machine Translation)

Markovian assumption that the next token only depends on the recent context (Markov, 1913). Based on this assumption, $p(w_k \mid w_1, \ldots, w_{k-1})$ is approximated, e.g., by $p(w_k \mid w_{k-2}, w_{k-1})$ in the case of a bi-gram model; $p(w_k \mid w_{k-2}, w_{k-1})$ is then approximated statistically based on the number of times $w_k$ appeared after $w_{k-2}$ in a large corpora of text, $w_{k-1}$ divided by the total number of times $w_{k-2}, w_{k-1}$ appeared in the corpora.

With the advent of word embeddings (Bengio et al., 2000; Mikolov et al., 2013), neural approaches to language modeling gained more popularity, in which a neural network is used to predict the next token probability. The use of powerful neural networks such as feedforward network (Bengio et al., 2000), recurrent neural model (Mikolov et al., 2012), specifically, long-short term memory (LSTM) models (Hochreiter & Schmidhuber, 1997) and Transformer models (Vaswani et al., 2017) allowed for predicting the next token probability based on a much longer and a variable length context, thus enabling better estimation of $p(w_k \mid w_1, \ldots, w_{k-1})$.

The increased power of neural LMs led to a new learning paradigm for NLP problems. Historically, the dominant learning paradigm for NLP problems was to train models on task-specific data from scratch. Consequently, for each new task, the model had to learn everything from scratch. This often resulted in poor generalization, especially in the cases where previously unobserved vocabulary was observed at the test time. In the subsequent paradigm, an LM was first pre-trained on a large corpora of text making it learn about how language works and gain a vast amount of knowledge about the world (Petroni et al., 2019; Lin et al., 2020; Sung et al., 2021; Yuan et al., 2023); the pre-trained LM (PLM) was then further finetuned on data from the new tasks (Sarzynska-Wawer et al., 2021; Devlin et al., 2018) thus teaching the general PLM the specifics of the new task. This paradigm often resulted in faster learning and higher predictive performance. It was later shown that further finetuning a PLM on multiple tasks leads to better transfer of knowledge across tasks and may lead to better performance on new tasks (Raffel et al., 2020).

## 2.1 In-Context Learning

As the scale of the PLMs and the scale of the datasets on which these models were pre-trained increased – leading to pre-trained Large Language Models (LLMs), it was discovered that pre-trained LLMs (hereafter, referred to as *LLMs* for brevity) have a remarkable capability of learning in-context from a few demonstrations (Brown et al., 2020). That is, LLMs were shown to be able to adapt to new tasks by only seeing a few examples of the new task in their input, as opposed to needing additional training data or fine-tuning. This is typically referred to as *few-shot in-context learning*. In the following, we will illustrate this formally.

Let $\mathcal{T}$ be a task and $q_* \sim \mathcal{T}$ represent a sample query from this task for which we would like to find an answer using an LLM. In the case of few-shot learning, we find or construct multiple demonstrations $\{d_1, \ldots, d_k\}$ where each demonstration $d_i = (q_i, a_i)$ contains a query $q_i \sim \mathcal{T}$ and the answer $a_i$ to that query, and feed an input of the form

> **Prompt 1: Few-shot Prompt**
>
> $q_1\ a_1\ \ldots\ q_k\ a_k\ q_*$

to the LLM. The input is typically referred to as *prompt*. It is common to add some separator tokens to the prompt so the boundaries of the demonstrations and the questions and answers within those demonstrations are clear. An example prompt will then be as follows:

> **Prompt 2: Prompt 1 + Separator Tokens**
>
> Demonstration 1: Query: $q_1$, Answer: $a_1$
> ...
> Demonstration k: Query: $q_k$, Answer: $a_k$
> Demonstration $k + 1$: Query: $q_*$, Answer:

The demonstrations serve as a few examples of the task helping the LLMs learn both the input and label space as well as the mapping from the inputs to the labels (Wei et al., 2023; Pan, 2023) both in context (without any weight updates), so a similar mapping can be applied to $q_*$. Few-shot learning is a remarkable capability of LLMs, showcasing their generalization ability to rapidly adapt to a wide range of tasks with explicit instruction. While LLMs show strong few-shot learning capabilities off-the-shelve, it has been shown that warming them up by finetuning them on few-shot data from multiple tasks will further boost their few-shot learning capability (Min et al., 2022a; Chen et al., 2022; Radford et al., 2019).

Another remarkable ICL capability of LLMs is to learn from in-context instructions: finetuning LLMs on instructions from multiple tasks makes them learn to follow instructions for new tasks (Ouyang et al., 2022; Longpre et al., 2023; Zhang et al., 2023). In this case, commonly known as *instruction tuning*, the LLM is finetuned on data of the type $I^{\mathcal{T}}, q^{\mathcal{T}}, a$ where $I^{\mathcal{T}}$ represents the instructions for a task $\mathcal{T}$ describing how the task should be performed, $q^{\mathcal{T}}$ represents a query from task $\mathcal{T}$ and $a$ represents the answer. The finetuning is performed on data from multiple tasks and multiple queries from each task. It is also possible to combine instructions with few-shot demonstrations, in which case an example prompt is as follows:

> **Prompt 3: Prompt 2 + Task Instructions**
>
> **[Task instructions]**
> **[Prompt 2]**

**Benefits of ICL:** Compared to the aforementioned approach of utilizing LLMs which involves pre-training followed by fine-tuning, ICL offers several key advantages. Firstly, fine-tuning may not always be feasible due to restricted access to the LLM, inadequate computational resources, or inadequately labeled data (Brown et al., 2020), whereas ICL requires fewer resources, less data, and is easier to serve through API calls. Additionally, ICL avoids the issues commonly associated with fine-tuning, such as overfitting or shocks (Ying, 2019; Kazemi et al., 2023a), as it does not modify the model's parameters, allowing it to remain general.

## 2.2 What Makes for Good Demonstrations?

Several works try to provide theoretical justifications and insights into how LLMs learn from a few in-context demonstrations (Xie et al., 2021; Garg et al., 2022; Von Oswald et al., 2023). However, the exact reasons behind this capability are still largely unclear making it difficult to select optimal few-shot demonstrations. Fortunately, various empirical results show the effect of the few-shot demonstrations on the predictive accuracy of the LLMs and provide suggestions on the best practices for preparing them. They also show the brittleness of the LLMs in the choice, format, and order of the few-shot demonstrations. Here, we describe some of the more prominent ones.

**Number of Demonstrations:** LLMs generally benefit from more demonstrations, but as the number of demonstrations increases the rate of improvement typically decreases (Brown et al., 2020; Ye et al., 2023b; Min et al., 2022b). Generation tasks have been shown to benefit from an increased number of demonstrations more than classification tasks (Li et al., 2023b). Toward increasing the number of demonstrations, one barrier is the maximum context size of the LLM. While the size of the context has been increasing over time with newer LLMs (Team et al., 2024; Chen et al.; Reid et al., 2024; Peng et al., 2023a; Gu & Dao, 2023; Li et al., 2024), it may still be problematic for datasets with long input texts or classification datasets with many classes.

**Demonstration Formatting:** Various works have shown that the formatting and wording of the prompts can play a crucial role in the performance of the LLM (Jiang et al., 2020; Shin et al., 2020; Kojima et al.; Yang et al., 2023). For example, Kojima et al. shows that simply adding *Let's think step by step* to the prompt makes LLMs reason step by step and solve substantially more problems, and Weller et al. (2023) show that adding *According to Wikipedia* to the prompt makes them more factual. Moreover, Min et al. (2022b) shows that besides the text formatting, the label space and the distribution of the input text in the demonstrations are also of immense importance.

**Order of Demonstrations:** The order of demonstrations has been shown to substantially affect the model performance. For example, Lu et al. (2022b) show that on some tasks, the model performance can range from near-random to state-of-the-art depending on the order of the prompts, and Zhao et al. (2021) show that answers appearing toward the end of the prompt are more likely to be predicted by the model.

**Diversity of Demonstrations:** Another important factor in the success of few-shot learning is the diversity of the demonstrations (Li et al., 2022b). Zhang et al. (2022b) propose to select a diverse set of questions as few-shot examples. Ma et al. (2023) propose a fairness metric for selecting demonstrations which encourages selecting diverse few-shot demonstrations that produce a near uniform predictive distribution for a semantic-free input. These works have shown that the diversity of demonstrations is crucial for achieving better in-context learning (ICL) performance.

**Chain of Thought (CoT):** It has been shown that including a rationale for the answer significantly improves model performance, especially for models that are larger than a certain size (Suzgun et al., 2022). The rationale is commonly known as *chain of thought (CoT)* (Wei et al., 2022). In the case of CoT prompting, the demonstrations are typically formatted as:

$$\text{Query}: q_i, \quad \text{Rationale}: r_i, \quad \text{Answer}: a_i$$

with the rationale appearing before the final answer. Naik et al. (2023) found that the diversity of reasoning path is important as well, therefore they propose *DiversePrompting* where an LLM is prompted to generate diverse reasoning path to solve a problem. Several works have investigated the reason behind the efficacy of CoT prompting and how to improve the prompts and rationales (Wang et al., 2022a; Lanham et al., 2023).

## 3 In-context Learning with Demonstration Retrieval

Traditionally, the same set of few-shot demonstrations is used on all queries, which can be suboptimal especially when there are high variations among the queries. An alternative is to *retrieve* few-shot demonstrations that

are tailored to the current query. Previous work has shown that demonstration retrieval leads to substantial improvements in the task metrics, compared to manually curated or randomly selected demonstrations (Luo et al., 2023; Ye et al., 2023a). Furthermore, LLMs have been shown to become less sensitive to the factors such as demonstration ordering (Section 2.2) when retrieved demonstrations are used (Li et al., 2023b).

This section gives an overview of the retrieval-based ICL (RetICL). We start by defining ICL with *retrieved demonstrations*. Formally, given a query $q_*$ and a **retrieval corpus** $\mathcal{C}$, a **demonstration retriever** $\mathcal{DR}$ selects a set of demonstrations $\{d_1, \ldots, d_k\} \sim \mathcal{C}$, where each demonstration is $d_i = (q_i, a_i)$. The LLM input sequence becomes $(d_1, \ldots, d_k, q_*)$. The goal of the retriever is to select demonstrations that maximize the probability of the correct answer $a_*$.

The success of RetICL depends on several factors. This section explores design choices, including the retrieval objectives, retrieval inference strategy, and retrieval corpus. Then in Sections 4 and 5, we explore the retriever models and how to train them to tailor to downstream tasks.

## 3.1 Retrieval Objectives: Similarity and Diversity

Various retrieval objectives for selecting and tailoring in-context examples for LLMs have been explored (Luo et al., 2023; Rubin et al., 2022; Ye et al., 2023a; Dalvi et al., 2022; Cheng et al., 2023; Li et al., 2023b). There are two primary retrieval objectives for selecting demonstrations: similarity and diversity. Similarity involves selecting demonstrations most akin to the query and can be based on language similarity (term matching (Luo et al., 2023; Rubin et al., 2022; Agrawal et al., 2022; Ye et al., 2023a; Dalvi et al., 2022) or semantic matching (Rubin et al., 2022; Li & Qiu, 2023b; Wang et al., 2023a; Liu et al., 2022)), structural aspects (sentence structure (Poesia et al., 2021; Levy et al., 2022; Drozdov et al., 2022), reasoning structure (Fu et al., 2022)). Most studies focus on language similarity, with fewer addressing structural similarity, often due to the challenges in extracting a query's structure in many tasks (Levy et al., 2022). Beyond similarity, some work has found that the diversity of demonstrations is important. The motivations for diversity include avoiding repetitive demonstrations (Zhang et al., 2022b), bringing different perspectives (Yu et al., 2023), and maximizing the demonstrations' coverage of the test query, in terms of covering either its words or syntactic structures (Levy et al., 2022). Measuring the diversity of multiple demonstrations is a major technical challenge. Ye et al. (2023a) applied determinantal point processes (DPP) a probabilistic model to measure the negative interaction (Kulesza et al., 2012), to measure the diversity. Levy et al. (2022) found that diversity and coverage are important when the model is unfamiliar with the output symbols space. It is noteworthy that researchers have found that ICL benefits more from demonstrations with higher complexity in some scenarios (Fu et al., 2022), where they define the complexity in terms of the query length or reasoning steps. However, Fu et al. (2022) employed heuristic rules to define complexity and pre-selected demonstrations accordingly. Their research revealed that using a similarity-based retriever led to improved performance in a specific mathematical reasoning task. This might indicate that combining similarity and complexity considerations could be a promising strategy for enhancing the approach to reasoning tasks.

## 3.2 Inference Strategy to Retrieve Few-shots Demonstrations

This section explores various strategies for employing a retriever to gather $k$ demonstrations. We divide these into three distinct methodologies.

**Top-k Retrieval** This is the simplest and most popular retrieval strategy (Liu et al., 2022; Rubin et al., 2022; Li et al., 2023b; Luo et al., 2023; Gao et al., 2023). To obtain $k$ demonstrations, given a query, the retriever ranks the demonstrations and then selects the top-$k$ demonstrations. Thus, each demonstration is chosen independently of the others. This method is straightforward and fast, however, it might not yield the best combination of $k$ demonstrations as these demonstrations might be homogeneous.

**Clustering Retrieval** To mitigate the issue of homogeneity in one-hot retrieval, clustering retrieval approaches (Li et al., 2022a; Zhang et al., 2022b; Li & Qiu, 2023b) categorize all demonstrations into $k$ sub-groups aiming to group similar demonstrations together. Then given a query, the retriever picks the most similar demonstration from each sub-group resulting in a final set of $k$ demonstrations. The core principle of

clustering is to select a diverse range of demonstrations. Most of the work use SBERT (Reimers & Gurevych, 2019) to encode the demonstrations (only the question or the entire demonstrations) and then apply $k$-means for clustering.

**Iterative Retrieval**  The earlier retrieval strategies acquire each demonstration independently. However, in iterative retrieval, a retriever selects demonstrations based on both the query and previously retrieved demonstrations (Khattab et al., 2022; Levy et al., 2022; Drozdov et al., 2022; Ye et al., 2023a; Scarlatos & Lan, 2023). This process starts with a single query, for which the retriever finds one best demonstration. The query is then augmented (e.g. combined with the demonstration) to retrieve the next demonstration. This step is iteratively executed $k$ times to gather $k$ demonstrations. The general idea is to select the demonstrations that can complement each other. An an example of a work from this categorym, Scarlatos & Lan (2023) train an LSTM retriever using a reinforcement learning framework. During the inference phase, the retriever processes the input query to select the best initial demonstration. It then generates a new query representation by integrating the query with prior demonstrations, specifically utilizing the hidden state representation from the LSTM model. This process of updating the query representation and obtaining subsequent demonstrations continues iteratively until $k$ demonstrations are retrieved.

### 3.3 Retrieval Corpus

The retrieval corpus forms a pool of demonstrations that the retriever can access. Using annotated data is one of the most straightforward ways to construct the retrieval corpus. This setting assumes that training data related to a task is available, and thus can be used as the retrieval corpus. Under this setting, there are three main ways to construct the corpus that we will discuss individually below.

**In-Domain**  In this setting, an in-domain training set, independently and identically distribution (IID) with the test queries, is available and serves as the retrieval corpus. Most existing work take the full training set as the corpus. However, to be more annotation efficient, Hongjin et al. (2022) uses only a subset $M$ of the training set $N$ which includes the most representative and diverse ones, where $|M| << |N|$. One question that remains unanswered from the work of Hongjin et al. (2022) is how the predictive performance is affected as a function of retrieving from a subset $M$ instead of the entire training set $N$. While there is no follow-up work to answer this question, the closest comparison we find is the results in Ye et al. (2023a) where a similar setup as Hongjin et al. (2022) is used except that they use the entire training set as the retrieval corpus, and report lower performance on the SST-5 dataset (compare the Figure 3 in Hongjin et al. (2022) and Table 3 in (Ye et al., 2023a)). While there might be other differences (e.g. the number of the demonstrations and the templates being used to do the inference) between the two setups that may affect the final performance, this comparison implies that retrieving from a carefully selected subset might have comparable results to retrieving from the entire training set.

**Mix-Domain**  The previous scenario has one individual retrieval corpus for different tasks. Assuming that we want to test model performance on two tasks, then in the in-domain setting, there will be two retrieval corpora separately. Furthermore, the in-domain setting assumes that the model has knowledge about which task the test question belongs to such that when it comes to the retrieval phase, it knows which corpus to select the demonstrations from. However, this assumption does not hold in several real-world applications of LLMs. In the mix-domain setting (Wang et al., 2023a), the retrieval corpus is constructed from the combination of all tasks. At the inference time, given a question, the retriever will retrieve demonstrations from this mixed corpus; the demonstrations can come from the same domain as the test question or from other tasks. However, the authors have not discussed whether the mix-domain approach is more beneficial than the single-domain approach. Li et al. (2023b) propose a unified retriever trained on a mixture of 40 tasks. This unified retriever can be used for different tasks during inference. The authors have compared the unified retriever with the single-domain retriever (Rubin et al., 2022) and demonstrated improved performance. It's worth noting that while the unified retriever is trained on mix-domain data, the actual selection of demonstrations during inference depends on the specific implementation.

**Cross-Domain** In this setting, IID human-annotated demonstrations are not available for the test queries, so one uses annotated demonstrations from other similar tasks but different domain (Cheng et al., 2023; Shi et al., 2022). The distinguish between the mix-domain and the cross-domain settings is that in the former setting, part of the corpus is IID. Shi et al. (2022) describes a scenario where the goal is to parse a Chinese query into SQL. However, the demonstrations are sourced from an English Text-to-SQL corpus, a domain with significantly more resources than the target domain. Shi et al. (2022) employs this high-resource data as the retrieval corpus. To adapt to the target domain during inference with a LLM, the target query is translated into the same language as the demonstrations. Nie et al. (2022) presents a similar approach but the retrieval corpus consists of multiple cross-domain datasets. The main benefit of this setting is to address low-resource training data issues or situations where no training data are available.

**Unlabelled Queries with Automatically Generated Answers** The previous three corpora all presuppose the availability of human-annotated data. However, this assumption may not hold in real-life scenarios, particularly in streaming settings where users can pose questions without any pre-annotated answers. Several studies (Zhang et al., 2022b; Li & Qiu, 2023b) have suggested using LLMs to generate answers for unlabeled data. They apply filtering techniques to determine the quality of these generated answers, adding only those examples with high-quality answers to the retrieval corpus. The most widely used filtering technique is based on self-consistency (Wang et al., 2022c). This approach involves prompting the language model to generate multiple chains of thought and answers, then selecting the most common answer as the final response.

**Free Form Corpus** Another approach to deal with the lack of human-annotated data for similar tasks is to create pseudo-demonstrations from unstructured text. Toward this goal, Lyu et al. (2022) utilized the Demix dataset (Gururangan et al., 2022), which is not tailored for any specific task. To generate pseudo-demonstrations, a retriever selects the top-k most relevant sentences from the dataset. Subsequently, arbitrary labels are attached to each sentence to form the examples. Such pseudo-demonstrations are beneficial compared to the zero-shot setting. The motivation behind using arbitrary labels comes from (Min et al., 2022b), who found that replacing gold labels with random labels only marginally hurts performance, suggesting that the input-output format is key. Li et al. (2022a) propose a synthetic question answering generation method to create QA pairs using the synthetic generated passages by an LLM. The authors also compared the synthetic generated data with the training data as demonstrations and found that the performance is comparable to using the training set, without requiring extensive human labor to collect the training data.

## 4 Off-the-shelf Demonstration Retrievers

To achieve the retrieval objectives outlined above, researchers have explored various types of demonstration retrievers Robertson et al. (2009); Reimers & Gurevych (2019); Liu et al. (2019); Yang et al. (2019); Ni et al. (2021); Santhanam et al. (2021). A typical demonstration retriever encodes examples from the retrieval corpus and the query into some vector representations, and then a similarity measure (e.g. cosine similarity) is calculated between candidate demonstration embeddings and the query embedding to locate the most related demonstrations Rubin et al. (2022); Li & Qiu (2023b); Wang et al. (2023a). Given the limited understanding of the underlying mechanism through which retrieved demonstrations enhance the performance of LLMs, initial research efforts focused on a heuristic evaluation of readily available retrievers for this task (Liu et al., 2022; Zhang et al., 2022b). Subsequent research endeavors explored the design and development of learning-based retrievers specifically customized for retrieving demonstrations (Luo et al., 2023; Cheng et al., 2023; Li et al., 2023b). This section reviews representative off-the-shelf models and we will discuss the learning-based models in Section 5.

**Term-based Similarity** BM25 (Robertson et al., 2009) is one of the most popular term-based scoring methods due to its simplicity and effectiveness in producing relevant results. It takes into account both term frequencies and document lengths. It has been empirically demonstrated in various works (Luo et al., 2023; Rubin et al., 2022; Agrawal et al., 2022; Ye et al., 2023a; Dalvi et al., 2022) that using BM25 to select similar examples as few-shots in ICL can help improve the performance of many LLM inference tasks. While BM25 has become a standard baseline model in the field, it is not without its limitations. Due to its heavy

reliance on term frequency and document length, this approach may overlook crucial aspects such as semantic meaning and sentence structure. Consequently, BM25 might fail to retrieve instances that are semantically similar to the input query but lack exact term matches (Guo et al., 2016; Xiong et al., 2017; Mitra et al., 2018). Another drawback is that BM25 lacks the capability for fine-tuning in downstream tasks, making it less competitive compared to neural models which can be fine-tuned and customized for specific downstream tasks.

**Sentence Embedding Similarity**    In this approach, queries and documents are encoded to the same dense embedding space using an off-the-shelf sentence embedding model, and then similarity scores (e.g. cosine similarity) are calculated to rank the most relevant documents for each query. A rich collection of sentence embedding methodologies exists in the literature (Reimers & Gurevych, 2019; Yang et al., 2019; Xue et al., 2020; Wang et al., 2022b). Sentence-BERT (SBERT) (Reimers & Gurevych, 2019) is a modification of the pretrained BERT network that uses siamese and triplet network structures to derive semantically meaningful sentence embeddings. The effectiveness of SBERT embeddings for demonstration retrieval has been investigated in several works (Rubin et al., 2022; Li & Qiu, 2023b; Wang et al., 2023a), and the results show that retrieving demonstrations based on SBERT embeddings often provides a boost in performance compared to zero-shot or random few-shot selection. In the KATE method (Liu et al., 2022), the authors studied using vanilla RoBERTa (Liu et al., 2019) and finetuned RoBERTa on NLI (Bowman et al., 2015b) and STS-B (Cer et al., 2017) datasets for selecting good demonstrations, and found that the finetuned version on task-related datasets offered further empirical gains. Note that here, the demonstration retriever is not trained for ICL demonstration retrieval based on task-specific data (a topic which we will discuss in Section 5); instead, the retriever is finetuned related tasks to provide a better notion of similarity for the task at hand. So we still categorize it as an off-the-shelf retriever. Shi et al. (2022) extends the use case to cross-lingual few-shot retrieval in the Text to-SQL semantic parsing task, and they use mSBERT (Reimers & Gurevych, 2019), mUSE (Yang et al., 2019) and mT5 (Xue et al., 2020) as the baseline models for comparison. Other widely used baseline models for demonstration retrieval include $E5_{\text{base}}$ (Wang et al., 2022b), SimCSE (Gao et al., 2021b). Instead of relying on "word matches" as in BM25, these sentence embedding similarity approaches can better capture semantic similarity (for example, synonyms, and related topics), however computationally they might be more expensive.

**Pretrained Dual Encoder**    Dual Encoder that are pretrained on information retrieval or question-answering tasks can grasp the intricate relationships between complex logical concepts and reasoning processes by employing different semantic embeddings for queries and candidates (Li & Qiu, 2023b). Most of the Dual Encoders use one global semantic representation for the query and candidates, such as GTR (Ni et al., 2021), while few others such as ColBERT (Santhanam et al., 2021) use token semantic representations. GTR (Ni et al., 2021) is a T5-based dual encoder model that is pretrained on the CommunityQA (Abujabal et al., 2019) and finetuned on the MS Marco dataset (Nguyen et al., 2016). ColBERTv2 is a state-of-art retrieval model that adopts the late interaction architecture (Khattab & Zaharia, 2020) and is trained on the MS Marco dataset. While publicly available pretrained dual-encoder retrievers are not specifically optimized for few-shot demonstrations retrieval tasks, they have been demonstrated to be successful in helping LLMs to learn from the selected examples. For instance, Luo et al. (2023) studied applying GTR (Ni et al., 2021) to select semantically similar examples as demonstrations, and empirically proved that this approach brought in better performance gain than random fewshots for both PaLM (Chowdhery et al., 2023) and FLAN (Chung et al., 2022) models. Another example is Khattab et al. (2022), who reported results for employing ColBERTv2 (Santhanam et al., 2021) as the retrieval module in their DEMONSTRATE–SEARCH–PREDICT (DSP) framework for ICL. In the proposed framework, it is used to retrieve both (i) related knowledge during the search stage and (2) top k similar examples as demonstrations. The pretrained models serve as strong retrievers for demonstrations even without fine-tuning on downstream tasks. Additionally, the pretrained weights can be utilized for fine-tuning the demonstration retriever, significantly reducing the amount of training data required compared to the pre-training phase, as we will show in §5.

# 5 Fine-tuned Demonstrations Retrievers

Although off-the-shelf retrievers have shown some promise in retrieving demonstrations for LLMs, the retrieved demonstrations given by the off-the-shelf retrievers might not represent the nature of the task and how the task should be solved in general. Therefore, it might lead to sub-optimal performance. Researchers thus have started to explore learning-based methods to further push the boundaries. A typical objective when designing a good demonstration retriever is: if an LLM finds a demonstration useful when being used as an illustrative example, the retriever should be encouraged to rank the demonstration higher. This allows us to train models directly relying on signals from query and output pairs in the task of interest, without human annotations. To develop a demonstration retriever, the majority of approaches utilize current dual encoder models (Karpukhin et al., 2020; Ni et al., 2021). The key variations lie in the methods of gathering training data and formulating training objectives. We will explore these aspects in more detail in the subsequent sections.

## 5.1 Collecting Training Data for Demonstration Retriever

**Based on LLMs Signals**  A popular approach to collecting training examples is to use the supervisory signals from LLMs. In this case, a typical paradigm is to first employ some filtering mechanisms (Cheng et al., 2023) or unsupervised retrievers (e.g. BM25 and SBERT) (Luo et al., 2023) as the initial retriever, this step can help limit the pool size for mining the right training data. Then a scoring LLM, which serves as a proxy for the inference LLM, is used to score each candidate demonstration $d$. Here the score is defined as $s(e) = p(a|d, q)$ which is the conditional probability of output answer $a$ given the input query $q$ and demonstration $d$. Another approach is to train a smaller reward model that can provide more fine-grained supervision for dense retrievers. For example, Wang et al. (2023a) proposed to finetune a cross-encoder model serving as a teacher model for training the retriever.

Once a score is obtained, a retriever can be trained that predicts these scores directly (Ye et al., 2023a). Alternatively, the candidate demonstrations can be ranked for each query based on their scores, considering the top-ranked demonstrations as *positive* examples that help the LLM get to the right answer and the bottom-ranked ones as *negative* examples that mislead the LLM towards the wrong answers; then a retriever can be trained which separates positive examples from negative examples (Rubin et al., 2022; Cheng et al., 2023; Luo et al., 2023).

There are different strategies for choosing the scoring LLM. Ideally, one uses the inference LLM itself as the scorer in order to perfectly reflect its preferences (Li et al., 2023b; Shi et al., 2022). However, training retrievers requires large amounts of labeled data, and it may be expensive use very large models for labeling. Consequently, for scoring one may gravitate towards utilizing smaller models, especially those within the same model family as the inference LLM (Luo et al., 2023; Cheng et al., 2023; Rubin et al., 2022).

**Model-Free**  One approach to collecting training data for demonstration retriever is to directly measure the similarity between the labels of the candidate demonstrations and the label of the query, and use this similarity as a proxy of the importance of a demonstration (Hu et al., 2022; Poesia et al., 2021). For instance, Hu et al. (2022) explored a dialogue context where labels are structured as a sequence of stages. The similarity between a query's label and a demonstration's label is determined by calculating the average F1 scores of these two labels. This method adopts a heuristic approach (i.e. stage changes), presuming that the similarity metric can closely resemble the preference for good demonstrations from an LLM, and it often necessitates domain-specific expertise for design.

## 5.2 Training Objectives

Thus far, we have explored the creation of training data for demonstration retrievers in the context of ICL. We now proceed to examine the commonly used loss functions for training retrievers.

**List-wise Ranking Loss**  The list-wise ranking approach looks at a list of candidate documents for a given query and tries to capture the correct ordering for it. Li et al. (2023b) proposed to inject the ranking signals

into the retriever using an approach inspired by LambdaRank (Burges, 2010). More formally, given each query $q$, they first rank all $l$ candidate documents according to their relevant scores $S = \{s(d_i)\}_{i=1}^{l}$, according to which the associated ranking $\mathcal{R} = \{r(d_i)\}_{i=1}^{l}$ is computed. Then the loss function is defined as follows:

$$\mathcal{L}_{\text{listwise}} = \sum_{d_i, d_j} \max\left(0, \frac{1}{r(d_i)} - \frac{1}{r(d_j)}\right) * \log(1 + e^{\text{sim}(q_i, d_j) - \text{sim}(q_i, d_i)})$$

where $\text{sim}(q, d)$ is the relavance between a candidate demonstration $d$ and the input $q$. In the list-wise ranking objective, retriever can benefit from the full ranking of the candidate set to make accurate predictions for the most relevant demonstrations. However, obtaining the full ranking list and calculating the loss function on top of it might be very expensive and time-consuming. Additionally, the model is trained to discern the relative preferences between examples without explicitly determining whether an example can serve as an absolute good demonstration.

**InfoNCE Loss** Another widely adopted training procedure is contrastive learning using the InfoNCE loss (Rubin et al., 2022; Cheng et al., 2023; Luo et al., 2023). When positive and negative examples can be correctly identified, InfoNCE loss is an effective loss function because it can take advantage of the supervisory labels to produce a representation that sets apart the useful examples for demonstration retrieval. In this approach, each training instance is given in the form of $< q_i, d_i^+, d_{i,1}^-, ... d_{i,k}^- >$. Here $d_i^+$ is a selected positive example concerning the input $q_i$, and the negative examples consist of one hard negative example $d_{i,1}^-$ and $k$ random examples from the other instances in the same mini-batch. Then the typical contrastive loss can be defined as

$$\mathcal{L}_{\text{cont}} = \mathcal{L}(q_i, d_i^+, d_{i,1}^-, ... d_{i,k}^-) = -\log \frac{e^{\text{sim}(q_i, d_i^+)}}{e^{\text{sim}(q_i, d_i^+)} + \sum_{j=1}^{k} e^{\text{sim}(q_i, d_{i,j}^-)}}$$

The random negative examples from the same mini-batch are called *in-batch negatives*. They are typically selected from both the positive examples and hard negative examples of other instances.

**Distillation by KL Divergence** Ye et al. (2023a) claims that although the InfoNCE loss has been found effective in training demonstration retrievers and can learn which examples might be superior to others, it has the same treatment for all negative examples and the predicted scores from LLM are not fully utilized. As an alternative to train a demonstration retriever using positive and negative examples, Shi et al. (2022) proposed to train the retriever by directly distilling the LLM's scoring function. More specifically, the retriever model is designed to produce ranking scores that match the usefulness of a demonstration to help with the LLM inference; this is done by minimizing the KL-divergence between the top $K$ examples score distribution from scoring LLM and the ranking score distribution produced by the retriever

$$\mathcal{L}_{\text{distill}} = \text{KL}(p_{\text{LLM}} || p_{\text{retriever}}) = \sum_{k=1}^{K} p_{\text{LLM}}(d_k) log\left(\frac{p_{\text{LLM}}(d_k)}{p_{\text{retriever}}(d_k)}\right)$$

**Multiple Objectives** The training signals, especially the LLM scores, are typically rich enough to define multiple training objectives that can be ensembled. For instance, Wang et al. (2023a) proposed to train the demonstration retriever model with combined objectives: (1) knowledge distillation from the trained reward model which can capture the preferences of LLMs over the retrieved candidates (2) InfoNCE-based contrastive loss to incorporate the in-batch negatives. More specifically, the resulting loss function is as follows:

$$\mathcal{L}_{\text{combined}} = \alpha \mathcal{L}_{\text{cont}} + \mathcal{L}_{\text{distill}}$$

Here $\alpha$ is a constant that controls the relative importance of the two losses. They claimed that with the multi-objective function, both the absolute scores and supervised signals are taken into consideration. Another example is Li et al. (2023b), who trains a universal retriever with both the list-wise ranking loss and InfoCNE loss. Both losses show additive effects on the model.

**Iterative Training**   Regarding training strategies, most research efforts have centered on fine-tuning a single retriever. Wang et al. (2023a) and Li et al. (2023b) instead proposed to iterate the retriever model multiple times. More specifically, the retriever trained in iteration $i$ will be employed to retrieve a new set of candidates for the subsequent iteration $i + 1$. Such an iterative training approach allows progressively improving retriever quality by mining better positive and hard negative examples at each iteration.

**Diversity Training**   The Determinantal Point Process model (Alex Kulesz, 2012) defines a probability distribution over all the combinations of candidate demonstrations, giving high probability to subsets that contain relevant and diverse items (Levy et al., 2022). It models diversity by incorporating cross-candidate similarity scores, and models similarity via a per-candidate relevance score, i.e., a similarity score between a candidate and the test query. In addition to using DPP directly (Levy et al., 2022), Ye et al. (2023a) also fine-tuned a DPP model and demonstrated meaningful improvements over pure similarity-based methods.

**Re-ranker Training**   It is not uncommon that people adopt a two-stage retriever-reranker architecture for ICL retrieval in the literature to further improve the exemplar selection process (Shi et al., 2022). Generally, a dual-encoder-based retriever can encode query and candidate documents for fast indexing and searching, but neglect the finer-grained token-level interactions. Cross-encoder-based reranker, on the other hand, can capture the subtle relationship but is time-consuming. We can benefit from both of these methods by chaining two methods together. In the first stage, a retriever model is used to quickly select the top $N$ examplers to limit the candidate pool of interest, then a reranker reranks the retrieved $N$ examplars and uses the top $K$ exemplars to construct a prompt. Sigmoid cross-entropy loss is typically used for training the reranker. Lu et al. (2022a) also utilizes a similar structure as the reranker to select the demonstrations from random candidates. The reranker is trained using reinforcement learning.

### 5.3   Summary

Here, we summarize the advantages and disadvantages of various retriever models. The off-the-shelf retrievers are easy to use without any downstream task finetuning and typically demonstrate stronger performance than random demonstrations. One exception is in commonsense reasoning tasks where Zhang et al. (2022b) and Ye et al. (2023a) found that for these tasks, random demonstrations are consistently better than retrieval-based method. Cheng et al. (2023) also show that retrieved demonstrations harm commonsense reasoning and coreference resolution tasks. Among the three categories of off-the-shelf retrievers, sparse retrievers such as BM25 are more index-efficient. This feature becomes particularly valuable when dealing with large volumes of demonstrations and limited hardware memory, making BM25 a preferable choice under such circumstances. In contrast, sentence-embedding similarity-based methods and dual-encoder-based retrieval systems, which are trained on language tasks, excel in capturing more semantically focused retrieval. Regarding performance, Luo et al. (2023) compared BM25 with dual encoder (GTR) across 5 tasks, and they found that the average performance of these two is very similar (within 0.5% difference), and BM25 outperformed the dual encoder in some tasks and vice versa. In another study, Ye et al. (2023a) observed a similar trend highlighting that no single retriever consistently outperforms others across different tasks. Both Rubin et al. (2022) and Li et al. (2023b) found that BM25 is better than SBERT on semantic parsing tasks, while Li et al. (2023b) found that SBERT is better than BM25 on sentiment analysis tasks.

Nevertheless, fine-tuned retrievers demonstrate superior performance compared to their off-the-shelf counterparts. The main drawback of fine-tuned retrievers lies in the high cost of obtaining training data. Additionally, the common practice of employing task-specific retrievers complicates the system and limits its generalizability, though there were recent works attempting to address this. For instance, Li et al. (2023b) trained a universal retriever that shows stronger performance than task-specific demonstration retriever on most of the tasks.

The choice of training objectives for fine-tuning a retriever depends on the specific goals, dataset characteristics, availability of labeled positive and negative examples, and computational constraints. For general retrieval tasks, when absolute relevance score is important, KL divergence is the ideal solution since it is optimized for distribution alignment. If there is a way to clearly differentiate positive examples from negative examples, InfoNCE is computationally cheap and works well, but as it does not emphasis absolute relevance, the ordering of the retrieved documents may be not as accurate. List-wise ranking loss optimizes the entire ranking order

directly, but is the most computationally expensive. With these trade-offs, there is no clear winner, so we often see in literature that empirical results serve as the primary guide for selecting which training objective to use, and ensembling training objectives proves to be effective (Wang et al., 2023a; Li et al., 2023b).

# 6 Applications

The effectiveness of retrieval-based ICL has been showed in four categories of tasks: 1) natural language understanding, 2) reasoning, 3) knowledge-based QA, and 4) Text generation. We discuss each category below.

Natural language understanding tasks that benefit from RetICL include sentiment analysis (SA) (Socher et al., 2013; Zhang et al., 2015; Go et al., 2009), paraphrase detection (PD) (Dolan et al., 2004; Zhang et al., 2019), reading comprehension (RC) (Rajpurkar et al., 2016; Khashabi et al., 2018; Clark et al., 2019; Khashabi et al., 2018; Clark et al., 2019; Mihaylov et al., 2018), and natural language inference (NLI) (Williams et al., 2018; Wang et al., 2018; Bowman et al., 2015a; De Marneffe et al., 2008). Specially, RetICL shows noticeable improvements on SA and NLI tasks (Liu et al., 2022; Ye et al., 2023a).

Reasoning tasks that benefit from RetICL include mathematical reasoning (Cobbe et al., 2021; Lu et al., 2022a; Ling et al., 2017), commonsense reasoning (CSR) (Talmor et al., 2019; Zellers et al., 2019; Bisk et al., 2020; Roemmele et al., 2011),and ethical Reasoning (Jiang et al., 2021). Such tasks are usually accompanied by CoT. Luo et al. (2023) demonstrate that retrieved demonstrations can be combined with the CoT technique to further enhance performance in mathematical reasoning tasks, showing that RetICL improves on top of CoT. Zhang et al. (2022b) emphasize the importance of diversity for these tasks. The iterative retrieval strategy, as noted by (Scarlatos & Lan, 2023), shows the most significant improvement in mathematical reasoning. Conversely, some studies have found that retrieval-based demonstrations perform worse than random demonstrations in commonsense reasoning tasks, such as CMSQA.

In Knowledge-based QA, external knowledge is required to answer the question (Berant et al., 2013; Kwiatkowski et al., 2019; Joshi et al., 2017; Clark et al., 2018). To tackle such tasks, the state-of-the-art systems usually retrieve relevant passages that might contain the answer to the question, and then feed such passages and questions together to a language model to generate the answer. Liu et al. (2022) shows that using retrieval-based ICL (sentence semantic similarity-based retriever with GPT-3) is almost comparable to a fine-tuned method. Ye et al. (2023a) shows that BM25 achieve $10+\%$ improvement on open-domain QA.

Text generation tasks such as code generation (CodeGen) (Zelle & Mooney, 1996; Lin et al., 2018), semantic parsing (SP) (Wolfson et al., 2020; Li et al., 2021; Andreas et al., 2020), text-to-SQL (Shi et al., 2022), Table-to-text (Table2Text) generation (Parikh et al., 2020), Data-to-Text (D2T) (Nan et al., 2021; Dušek et al., 2019) benefit from RetICL (Poesia et al., 2021; Rubin et al., 2022; Hu et al., 2022; Hongjin et al., 2022; Shi et al., 2022; Ye et al., 2023a; Agrawal et al., 2023). Rubin et al. (2022) shows that the retrieved demonstrations significantly outperform random demonstrations (e.g. BM25 is $25+\%$ better than random, and EPR is $30\%$ better than random).

Apart from different types of tasks, Hongjin et al. (2022) shows that in scenarios with limited training data, RetICL outperforms fine-tuning a model on such sparse data. Furthermore, leveraging data from a high-resource domain can enhance performance in a low-resource domain, as seen in cross-lingual contexts (Shi et al., 2022; Nie et al., 2022; Cheng et al., 2023).

# 7 Discussion

## 7.1 Guidelines for Choosing the Methods

Based on the surveyed works, we summarize the following guidelines for choosing a RetICL recipe.

- **Similarity and diversity:** While simple similarity-based retrieval is often sufficient, if the retrieval corpus contains many demonstrations that are similar to each other, the retrieved demonstrations may become homogeneous. If the demonstrations can be binned into sub-groups, one could use clustering

retrieval (Li et al., 2022a; Zhang et al., 2022b; Li & Qiu, 2023b). Alternatively, detrimental point processes (DPP) can be used if the distance between two demonstrations can be easily computed (Ye et al., 2023a; Levy et al., 2022).

- **Number of demonstrations:** As mentioned in Section 2.2, LLMs benefits from having more demonstrations, but has a diminishing return. To get the most gain, one can simply add as many demonstrations as the LLM context size allows. All retrieval strategies in Section 3.2 can select a dynamic number of examples (e.g. by over-retrieving and removing ones that overflow the context size).

- **Retrieval corpus content:** The retrieval corpus should contain demonstrations that are close to the target task. While some works found success at using mix-domain or cross-domain demonstrations (Section 3.3), the retrieval corpus still often contains demonstrations that are relatively similar to the target task; e.g. examples from different datasets of a similar task (Wang et al., 2023a) or different languages of the same task (Shi et al., 2022).

- **Correctness of the demonstrations:** LLM of different sizes learn from the demonstrations in different ways: small LLMs use the demonstrations for task recognition and rely more on its pretraining prior for deriving the answer, while large LLMs can learn the label mappings from the demonstrations (Kossen et al., 2024; Wei et al., 2023). It is thus more important to ensure the correctness of the demonstrations when using large LLMs.

- **Demonstration retriever:** As mentioned in Section 5.3, there is no clear consensus on the best off-the-shelf retriever. One recommendation would be to try BM25, which is simple but highly generalizable, as well as the most convenient model-based retriever such as SBERT or GTR. For a retriever trained specifically for demonstrations, one could first try the task-generic demonstration retriever (Li et al., 2023b).

## 7.2 Future Directions

**Retrieve Demonstrations From Raw Text** Much research assumes the availability of annotated samples that can be utilized as a retrieval corpus. Yet, when faced with a novel task, it is often the case that no such training dataset exists. While there are preliminary efforts to create pseudo demonstrations from open-ended corpora like Wikipedia (Lyu et al., 2022), the proposed method is restricted to classification tasks and the label to the demonstrations are randomly assigned. A potential approach to obtain pseudo demonstrations for generation tasks is the approach of Wan et al. (2023), where they assume a set of unlabelled queries available (without ground truth labels), and use LLMs to generate chain-of-thoughts and answers and then apply self-consistency (Wang et al., 2022c) to select high-quality demonstrations to form pseudo demonstrations pool. Employing this method of generating answers with sentences retrieved from a free-form corpus could potentially create high-quality pseudo demonstrations.

**Demonstration Retriever Design** Current demonstration retriever architectures are not significantly different from those used for raw text, with most efforts focusing on constructing the downstream training data. An open question remains: "How can retrievers be designed specifically for demonstrations rather than raw text?" A potential direction could involve training a task-aware retriever with instructions Asai et al. (2023), such that the fine-tuned retriever learns to tailor itself to instructional content. The instruction retriever should not only understand the semantic instructions but also retrieve the most relevant demonstrations from the database that follow these instructions.

**Retriever Training Methods** In Section 5, we explore various methods for training a retriever to search demonstrations. These methods largely depend on using an LLM to identify positive and negative demonstrations for a given question. This approach, while innovative, comes with significant computational demands. Moreover, the ambiguity in choosing what constitutes a positive or negative demonstration raises concerns about the quality of the training data for the retriever (Hashimoto et al., 2023). Addressing these challenges is crucial for the development of more efficient and reliable retriever training methods.

**Active Demonstration Retrieval**   Much of the current research is based on a static framework where the retrieval corpus remains constant. In practical situations, input distributions may change over time and models may come across new instances. In these cases, one may like to keep updating the retrieval corpus based on the new incoming queries, but since the labels for the new samples may not be available, a selection strategy is needed to select a representative subset of the incoming examples for annotation so they can be added to the retrieval corpus. This problem is reminiscent of the well-studied active learning problem. Zhang et al. (2022a) have studied how to actively select examples with unlabelled data, however, this study is not based on the retrieval setting. The combination of active learning and RetICL can be an interesting future direction.

**Retrieved Demonstrations for Small LM**   Most existing work focuses on large language models (LLMs) with more than 100 billion parameters, which are often used as off-the-shelf models without adaptation to downstream tasks. In contrast, small LMs not only offer the advantage of inference efficiency but also are easier to fine-tune compared to LLMs. This presents an opportunity to optimize both the retriever and the LM simultaneously, similar to end-to-end retrieval-augmented generation training methods (Guu et al., 2020; Lewis et al., 2020). By tailoring retrievers to produce demonstrations that align closely with the capabilities of small LMs, and fine-tuning these small LMs to adapt to in-context learning tasks, we can potentially enhance their performance on specific tasks.

**Theoretical Understanding of Why Are Similar Demonstrations Better Demonstrations?**   While extensive research has demonstrated that similar demonstrations are more effective than random ones in in-context learning (ICL), the underlying reasons for this effectiveness remain unclear. A parallel line of research focuses on understanding the mechanisms behind ICL's success. Several hypotheses have been proposed, including: Bayesian inference (e.g. locate previous learned concepts) (Xie et al.), inductive heads (Olsson et al., 2022), implicit gradient descent (Von Oswald et al., 2023; Dai et al.). These research directions may provide the foundation for explaining why similar demonstrations outperform random ones in ICL scenarios.

**Many-Shot Prompting**   With the rapid increase in the context length of the frontier LLMs (Reid et al., 2024; Anthropic, 2024), a new ICL paradigm has emerged where instead of providing only few demonstrations to the models, one can provide many shots. It has been observed that this approach can provide substantial boost in performance (Agarwal et al., 2024; Jiang et al., 2024; Anil et al., 2024), and also alleviate some of the issues with few-shot ICL such as sensitivity to the order (Bertsch et al., 2024). However, the increased context size in many-shot ICL introduces an extra inference cost. An interesting area for future research is to find ways of combining many-shot ICL and RetICL to get similar predictive performance with a lower inference cost.

**Fine-tuning For RetICL Few-shot Learning**   Most of the existing research utilizes a frozen LLM as the few-shot learner in the RetICL framework. As demonstrated by Gao et al. (2021a) and Izacard et al. (2023), fine-tuning LLMs on a few-shot learning task can be a promising approach to enhancing LLM performance during inference. An intriguing avenue could involve adapting the fine-tuning strategy from open-domain question answering models to RetICL (Lewis et al., 2020; Guu et al., 2020).

**RetICL for Vision and Language Models**   Vision and language models (VL) have demonstrated proficiency as few-shot learners (Alayrac et al., 2022; Awadalla et al., 2023; Li et al., 2023a), and researchers have integrated retrieval augmented generation methods with VL models (Luo et al., 2021; Gao et al., 2022; Yasunaga et al., 2023). In lieu of employing random demonstrations, Yang et al. (2024) leverages a retriever to select demonstrations for image captioning tasks. Additionally, Peng et al. (2023b) propose an ICD-LM to generate demonstrations for a given query, with each demonstration being represented by a token. The effectiveness of this method is evidenced by its performance on image captioning and vision question answering tasks. Despite the increasing significance of vision and language generation models in real-world applications, research on RetICL for VL models remains relatively underexplored.

**Many-Shot ICL:**   With the rapid increase in the context length of the frontier LLMs (Reid et al., 2024; Anthropic, 2024), a new ICL paradigm has emerged where instead of providing only few demonstrations to

the models, one can provide many shots. It has been observed that this approach can provide substantial boost in performance (Agarwal et al., 2024; Jiang et al., 2024; Anil et al., 2024), and also alleviate some of the issues with few-shot ICL such as sensitivity to the order (Bertsch et al., 2024). However, the increased context size in many-shot ICL introduces an extra inference cost. An interesting area for future research is to find ways of combining many-shot ICL and RetICL to get similar predictive performance with a lower inference cost.

## 8 Conclusion

This survey concentrates on few-shots In-Context Learning (ICL) using retrieved examples for large language models, a key aspect of Retrieval-Augmented Generation (RAG). We outline various retrieval strategies, diverse retrieval models, retrieval pools, techniques for training demonstration retrievers, and applications. Based on the comprehensive understanding of current trends, we suggest several promising future paths for enhancing the efficacy and functionality of this approach.

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
