# OpenReview forum: "In-context Learning with Retrieved Demonstrations for Language Models: A Survey"
_TMLR — Accepted by TMLR_

### Review · Reviewer_8UR5 · 2024-05-14

**Summary Of Contributions:**

This paper summarizes how to retrieve examples for in-context learning. It starts by reviewing in-context learning and demonstration retrieval, summarizing the main approaches in retrieval algorithms, and what tasks one would apply this framework to in practice.

**Audience:**

Yes

**Claims And Evidence:**

Yes

**Requested Changes:**

1. [Major] I hope for a better guide on which method to pick for practical use cases and how different losses on fine-tuning affect the downstream model. In my opinion, this would be the main practical relevance of this paper.
2. The following works might help contextualize the effect of demonstrations on in-context learning: https://arxiv.org/abs/2303.03846, https://arxiv.org/abs/2305.09731
3. Though more recent than your submission, these works discuss using larger context windows to fit more in-context learning examples which would remove the need for selection of examples: https://cdn.sanity.io/files/4zrzovbb/website/af5633c94ed2beb282f6a53c595eb437e8e7b630.pdf, https://arxiv.org/abs/2405.00200
4. General formatting errors throughout the document. For example, SBERT citation in Section 3.2, numbering of application domains at the start of Section 6...

**Strengths And Weaknesses:**

Strengths
- The paper is quite comprehensive, covering a lot of papers within this space
- The paper is well organized, allowing for a reader to look at what aspects are important to them

Weaknesses
- I find survey papers most helpful when they can synthesize viewpoints about the field and consider the connections between different works. This paper gave great summaries of all the contributions, but I do not feel like I was left with a greater sense of understanding of how all the works interact with each other. This is partially addressed in places like Section 5.3, but in my opinion, this could be a bigger emphasis.
- I believe this paper would be useful to a practitioner who would like to know the methods used for in-context demonstration retrieval. Its harder for me to see which communities outside of this group would benefit from the aggregation of summaries of methods in the current manuscript.

---

> ### Author Response · Authors · 2024-07-05
> **Response to Reviewer 8UR5**
>
> We appreciate your feedback and we have reposed to every point and the new manuscript reflected your suggestion. Particularly, our response are given in the following.
>
> 1.	[Major] Thank you for your comments and suggestions. We indeed want the paper to be useful for general NLP practitioners who use in-context learning. We have added a section (7.1 Guidelines for Choosing the Methods) that gives guidelines on the modeling choices to use, based on the works surveyed in the paper. As for how different fine-tuning losses affect the model, based on the works that linearly combine multiple losses ([Wang et al., 2023](https://aclanthology.org/2024.eacl-long.105/) and [Li et al., 2023](https://aclanthology.org/2023.acl-long.256/)), all of the losses have additive effects. We have added this comment to the paper in Section 5.2 (Multiple Objectives paragraph). We also added comparisons between fine-tuning losses in Section 5.3.
>
> 2.	Thank you for the extra references. We discussed the two references for contextualizing the effect of demonstrations in section 2.1. The new text reads as follows: “The demonstrations serve as a few examples of the task helping the LLMs learn both the input and label space as well as the mapping from the inputs to the labels \citep{wei2023larger,pan2023context} both in context (without any weight updates), so a similar mapping can be  applied to $q_*$. …”
>
> 3.	We  added a paragraph (Many-Shot Prompting) to our “Discussion and Future Direction” section referencing several many-shot ICL papers (including the ones you suggested) and explained the possible interplay between many-shot ICL and RetICL. The paragraph reads as follows: “With the rapid increase in the context length of the frontier LLMs \citep{reid2024gemini,anthropic2024claude}, a new ICL paradigm has emerged where instead of providing only few demonstrations to the models, one can provide many shots. It has been observed that this approach can provide substantial boost in performance \citep{agarwal2024many,jiang2024many,anil2024many}, and also alleviate some of the issues with few-shot ICL such as sensitivity to the order \citep{bertsch2024context}. However, the increased context size in many-shot ICL introduces an extra inference cost. An interesting area for future research is to find ways of combining many-shot ICL and RetICL to get similar predictive performance with a lower inference cost.”
>
> 4.	We fixed this citation format error and added the number in section 6.
>
> 5.	Thank you for pointing the typo, and they have been fixed in the new manuscript.

---

> > ### Comment · Reviewer_8UR5 · 2024-07-09
> >
> > Thank you, I appreciate the improvements and believe the paper will be more useful for practitioners now. I recommend acceptance.

---

> > > ### Author Response · Authors · 2024-08-05
> > >
> > > We really appreciate your insights and suggestions, which makes our work stronger.

---

### Review · Reviewer_m6ZC · 2024-06-05

**Summary Of Contributions:**

This survey focuses on retrieval-augmented in-context learning (ICL) in large language models, where demonstrations are retrieved from some existing dataset in order to improve the ICL process. The survey categorizes and describes a number of related work along axes such as choice of retrieval dataset or choice of retriever. It then makes some suggestions for future work in the RetICL area.

**Audience:**

Yes

**Broader Impact Concerns:**

I don't have any particular concerns.

**Claims And Evidence:**

No

**Requested Changes:**

Please see Weaknesses above for specific comments. The most critical points for acceptance are the additional citations to fill in many parts of the writing, as well as additional analysis, especially in the final section.

**Strengths And Weaknesses:**

## Strengths:
- ICL is a fast-growing topic, so this survey is quite timely.
- The categorization is clear, with works cited falling along the lines drawn by the paper.

## Weaknesses:

I will go into detail in the points below. At a high level I believe that this survey makes a number of claims throughout the text that are not well-substantiated by citations, which makes it a less than ideal entry point to the field. In addition I'm not totally certain why it makes sense to focus on such a specific topic as RetICL, though this isn't major to my review as the TMLR guidelines suggest against judging significance.

**Missing citations:** often, I feel that many claims are stated either without justification, or justified much later in the text far from the original claim. Examples:
  - "Furthermore, ICL avoids common issues with fine-tuning, such as overfitting" as far as I understand the works cited after this claim just say that fine-tuning has a tendency to overfit, not that ICL fixes the issue
  - "The effectiveness of such demonstrations is influenced by factors such as the quality, quantity, and ordering of the demonstrations" this is definitely discussed later, but I believe something should be cited at this point
  - "This method has not only consistently outshined approaches relying on... influencing factors" same as above this should probably have citations given it is such a strong claim and we the readers do not yet know much about the RetICL subfield
  - In the beginning of Section 2, "Language models (LMs)" could probably have more references, I think the goal is to be comprehensive, but I am willing to accept another perspective here
  - "Early LMs, which were mostly based on N-gram models... recent context" to cite only one object here, which is a survey chapter itself, doesn't seem right to me. I know that this topic is somewhat tangential to the main topic of the survey, but it seems like the history of the development is missing.
  - Similar to the above, after "neural approaches to language modeling gained more popularity" there are again only two citations for architectures separated by 20 years. There was a lot of development during that time and it seems a bit wrong for the text to ignore that.
  - "Few-shot learning is a remarkable capability... not trained on such data" I believe this really is a claim, and as such should be substantiated, though it is difficult to do so without looking at the entire dataset. The reason I disagree with this point is that it's possible that there are significant portions of the internet devoted to these repetitive demonstrations (for example math word problems), so in particular if an LLM is tested on math word problems then the capability is in-domain. I would think rather the power of few-shot learning is not that it is out of distribution per se, but rather that it applies quite generally.
  - "While the size of the context has been increasing over time with newer LLMS" it would be nice to have a citation here.
  - "Similarity involves selecting demonstrations... language similarity... structural aspects... or other criteria" these are described in detail later, but to be comprehensive it would be good to include the citations here.
  - the "Iterative Retrieval" heading relies entirely on Scarlatos and Lan, 2023, are there no other citations that the text might reference?
  - "To achieve... demonstration retrievers." cite here
  - "between candidate demonstration embeddings... demonstrations" cite here
  - "initial research efforts focused on a heuristic... for this task" cite here
  - "Due to its sole reliance on term frequency... certain instances" citations needed
  - "A rich collection... in the literature." citations needed here
  - "Text generation tasks that benefit... Data-to-Text." citations needed here to say that RetICL actually helps, beyond just citations for the tasks. The entire point of this survey is to promote and explain RetICL, so there should be ample evidence for that.
  - "There are some hypotheses... implicit gradient descent" this requires citations for the hypotheses and specific papers.

**Missing analysis:** There also seems to be a missed opportunity to make a more specific analysis in many cases:
- "While there might be other differences... training set" I think this is a little vague, it would be nice if the text exposed the actual differences between the two works and then attempted to make a more specific suggestion
- Under "Mix-Domain": "In the mix domain setting... from other tasks" the text does not present any analysis as to whether this mix-domain retrieval helps or hurts. It would be useful if such discussion were included.
- "Cross-domain": similar to above, the end of the paragraph does not discuss whether the referenced methods are helpful or hurtful, or what specific drawbacks/benefits are, it just mentions their existence.
- "Free Form Corpus": similar to the above two points, it would be nice to have some discussion on the tradeoffs between settings that are seen in the actual cited works. It's unclear what the effect of these different settings are, and the survey should expose some specific points.
- I believe the discussions in Section 7 make suggestions that are much too general. In particular "Is there a potential... tasks?" seems very straightforward to answer no for the simple reason that one might write rules for very closed domains. I would hope that there would be more insight on how the retriever might need to be designed specifically around *demonstrations* rather than raw text.
- Similar to above, "Examining and improving the ICL capabilities... direction to explore" does not have any suggestions or opinions on what is missing, which I might hope that such a survey would convey.
- "Such tasks are usually accompanied by CoT... reasoning task." This makes it seem like it is hard to disambiguate the effect of retrieval, if so what conclusion can we really draw from the importance of RetICL?
- At the end of the paragraph labeled "Pretrained Dual Encoder" it would be nice again to include discussion on the practical benefits/drawbacks of these specific encoders for RetICL, or something closer to the eventual setting.
- In the "Diversity of Demonstrations" heading, the methods are presented without discussion as to the benefit or drawback of the approaches, so I don't understand if this is a line worth pursuing or not.

**Other points:**
- Table 1 takes up more than a page, but I don't really understand how it is helpful for distilling the information. It presents a categorization, but because the organization is chronological it's not really clear what should be compared between methods. One option would be to carve up this table into the sections that are later described in the text, another might be to have separate tables for separate subsections. It would be nice if the caption were to highlight any specific trends we the readers are supposed to take.
- "One-hoc Retrieval" this is a strange way to phrase? Perhaps "One-step" or "Independent" might describe what the paper is aiming for? In addition, this heading contains no references, it might just be nice to have a few for comparison with later headings
- "Naik et al. 2023 propose... solving the problem" I do not understand what this particular line means. It would be helpful to rewrite somehow, though as I don't understand the intention I'm not sure what I can say constructively.
- In Section 2.1 it would be nice to number the centered demonstrations of prompt techniques. Then the 3rd set can reference the 2nd directly. It would also be helpful to bold "[Task Instructions]." I missed it initially on first glance and did not understand why both were included.

---

> ### Author Response · Authors · 2024-07-05
> **Response to Reviewer m6ZC (Missing Citation part 1)**
>
> We really appreciate your time on reading our manuscript and the detailed suggestions on improving the previous manuscript. Your suggestions have all been carefully considered and reflected on the new manuscript. In the following, we specifically mentioned our edition.
>
> Missing citation:
>
> 1.	We change this sentence to the following to prevent the confusion: Fine-tuning suffers from overfitting issue (Ying, 2019; Kazemi et al., 2023a), on the other hand, ICL does not have this issue since the weights of the model do not change.
> 2.	We add the citations at the end of this sentence. Brownet al. (2020); Naik et al. (2023); Kojima et al.; Lu et al. (2022b)
> 3.	This method has not only consistently outshined approaches relying on random or static hand-crafted demonstrations (Liu et al., 2022; Rubin et al., 2022; Luo et al., 2023) but has also demonstrated a remarkable resilience to a variety of influencing factors (Li et al.,2023b)
> 4.	Earlier LMs were mostly based on N-gram models (Jurafsky, 2021) and has been used in computing approximations to English word sequences (Shannon, 1948) and speech recognition system (Baker, 1990;Jelinek et al., 1975; Baker, 1975; Bahl et al., 1983; Jelinek, 1990). N-gram models are based on the Markovian assumption that the next token only depends on the recent context (Markov, 1913).
> 5.	You are right that there is lots of work that has been done in this scope, and it is hard to have a comprehensive citation. Therefore, here we want to focus on the original proposal of the model architectures or their application in the language model task.  We further change the previous sentence in the following way: The use of powerful neural networks such as feedforward network (Bengio et al., 2000), recurrent neural model (Mikolov et al., 2012), specifically, long-short term memory (LSTM) models (Hochreiter & Schmidhuber,1997) and Transformer models (Vaswani et al., 2017) allowed for predicting the next token probability based on a much longer and a variable length context, thus enabling better estimation of p(wk | w1, . . . , wk−1).
> 6.	Thank you for this insightful comment. You raise an excellent point about the potential for some few-shot learning tasks to be in-domain for LLMs, given the breadth of their training data. I agree that the true power of few-shot learning lies in its general applicability rather than it being strictly out-of-distribution. I appreciate this nuance and will revise my statement to reflect this more accurate understanding." Revised statement: "Few-shot learning is a remarkable capability of LLMs, showcasing their ability to rapidly adapt to a wide range of tasks with minimal explicit instruction.”
> 7.	While the size of the context has been increasing over time with newer LLMs (Team et al., 2024; Chen et al.; Reid et al., 2024; Peng et al., 2023a; Gu & Dao, 2023; Li et al., 2024)
> 8.	Similarity involves selecting demonstrations most akin to the query and can be based on language similarity (term matching (Luo et al., 2023; Rubin et al., 2022; Agrawal et al., 2022; Ye et al., 2023a; Dalvi et al., 2022) or semantic matching (Rubin et al., 2022; Li & Qiu, 2023b; Wang et al., 2023a; Liu et al., 2022)), structural aspects (sentence structure (Poesia et al., 2021; Levy et al., 2022; Drozdov et al., 2022), reasoning structure (Fu et al., 2022)).
> 9.	The earlier retrieval strategies acquire each demonstration independently. However, in iterative retrieval, a retriever selects demonstrations based on both the query and previously retrieved demonstrations Khattab et al. (2022); Levy et al. (2022); Drozdov et al. (2022); Ye et al. (2023a); Scarlatos & Lan (2023).
> 10.	To achieve the retrieval objectives outlined above, researchers have explored various types of demonstration retrievers Robertson et al. (2009); Reimers & Gurevych (2019a); Liu et al. (2019); Yang et al. (2019); Ni et al. (2021); Santhanam et al. (2021).

---

> ### Author Response · Authors · 2024-07-05
> **Response to Reviewer m6ZC (Missing Citation part 2)**
>
> 11. A typical demonstration retriever encodes examples from the retrieval corpus and the query into some vector representations, and then a similarity measure (e.g. cosine similarity) is calculated between candidate demonstration embeddings and the query embedding to locate the most related demonstrations Rubin et al.(2022); Li & Qiu (2023b); Wang et al. (2023a).
> 12. Given the limited understanding of the underlying mechanism through which retrieved demonstrations enhance the performance of LLMs, initial research efforts focused on a heuristic evaluation of readily available retrievers for this task (Liu et al., 2022; Zhang et al., 2022b). Subsequent research endeavors explored the design and development of learning-based retrievers specifically customized for retrieving demonstrations (Luo et al., 2023; Cheng et al., 2023; Li et al., 2023b).
> 13. We have rephrased a bit of the original sentence and add citations for the limitation of BM25. Due to its heavy reliance on term frequency and document length, this approach may overlook crucial aspects such as semantic meaning and sentence structure. Consequently, BM25 might fail to retrieve instances that are semantically similar to the input query but lack exact term matches (Guo et al., 2016; Xiong et al., 2017; Mitra et al., 2018).
> 14. A rich collection of sentence embedding methodologies exists in the literature (Reimers & Gurevych, 2019; Yang et al., 2019; Xue et al., 2020; Wang et al., 2022b).
> 15. Text generation tasks such as code generation (CodeGen) (Zelle & Mooney, 1996; Lin et al., 2018), semantic parsing (SP) (Wolfson et al., 2020; Li et al., 2021; Andreas et al., 2020), text-to-SQL (Shi et al., 2022), Table-to-text (Table2Text) generation (Parikh et al., 2020), Data-to-Text (D2T) (Nan et al., 2021; Dušek et al., 2019) benefit from RetICL (Poesia et al., 2021; Rubin et al., 2022; Hu et al., 2022; Hongjin et al., 2022; Shi et al., 2022; Ye et al., 2023a; Agrawal et al., 2023).
> 16. A parallel line of research focuses on understanding the mechanisms behind ICL’s success. Several hypotheses have been proposed, including: Bayesian inference (e.g. locate previous learned concepts) (Xie et al.), inductive heads (Olsson et al., 2022), implicit gradient descent (Von Oswald et al., 2023; Dai et al.).

---

> > ### Comment · Reviewer_m6ZC · 2024-07-16
> > **Response to 11-16**
> >
> > This all looks good to me.

---

> ### Author Response · Authors · 2024-07-05
> **Response to Reviewer m6ZC (Missing analysis)**
>
> In the following, we address the comment on missing analysis part.
>
> 1.	While there might be other differences (e.g. the number of the demonstrations and the templates being used to do the inference) between the two setups that may affect  the final performance, this comparison implies that retrieving from a carefully selected subset might have comparable results to retrieving from the entire training set
> 2.	In the mix-domain setting (Wang et al., 2023a), the retrieval corpus is constructed from the combination of all tasks. At the inference time, given a question, the retriever will retrieve demonstrations from this mixed corpus; the demonstrations can come from the same domain as the test question or from other tasks. However, the authors have not discussed whether the mix-domain approach is more beneficial than the single-domain approach. Li et al. (2023b) propose a unified retriever trained on a mixture of 40 tasks. This unified retriever can be used for different tasks during inference. The authors have compared the unified retriever with the single-domain retriever (Rubin et al., 2022) and demonstrated improved performance. It’s worth noting that while the unified retriever is trained on mix-domain data, the actual selection of demonstrations during inference depends on the specific implementation.
> 3.	We have added this sentence in the end to highlight the benefits of cross-domain setting, “The main benefit of this setting is to address low-resource training data issues or situations where no training data are available.” The two works we mentioned do not compare with in-domain or mix-domain since the datasets that they study do not have training data available.
> 4.	Another approach to deal with the lack of human-annotated data for similar tasks is to create pseudo-demonstrations from unstructured text. Toward this goal, Lyu et al. (2022) utilized the Demix dataset (Gururangan et al., 2022), which is not tailored for any specific task. To generate pusedo-demonstrations, a retriever selects the top-k most relevant sentences from the dataset. Subsequently, arbitrary labels are attached to each sentence to form the examples. Such pseudo-demonstrations are beneficial compared to the zero-shot setting. The motivation behind using arbitrary labels comes from (Min et al., 2022b), who found that replacing gold labels with random labels only marginally hurts performance, suggesting that the input-output format is key. Li et al. (2022) propose a synthetic question answering generation. method to create QA pairs using the synthetic generated passages by an LLM. The authors also compared the synthetic generated data with the training data as demonstrations and found that the performance is comparable to using the training set, without requiring extensive human labor to collect the training data
> 5.	Demonstration Retriever Design: Current demonstration retriever architectures are not significantly different from those used for raw text, with most efforts focusing on constructing the down-stream training data. An open question remains: “How can retrievers be designed specifically for demonstrations rather than raw text?” Demonstrations often require retrievers to account for instructional content and task understanding, which may demand more sophisticated and tailored approaches.
> 6.	Retrieved Demonstrations for Small LM Most existing work focuses on large language models (LLMs) with more than 100 billion parameters, which are often used as off-the-shelf models without adaptation to downstream tasks. In contrast, small LMs not only offer the advantage of inference efficiency but also are easier to fine-tune compared to LLMs. This presents an opportunity to optimize both the retriever and the LM simultaneously, similar to end-to-end retrieval-augmented generation training methods (Guu et al., 2020; Lewis et al., 2020). By tailoring retrievers to produce demonstrations that align closely with the capabilities of small LMs, and fine-tuning these small LMs to adapt to in-context learning tasks, we can potentially enhance their performance on specific tasks.
> 7.	Such tasks are usually accompanied by CoT. Luo et al. (2023) demonstrate that retrieved demonstrations can be combined with the CoT technique to further enhance performance in mathematical reasoning tasks.
> 8.	Add the following sentence: The pretrained models serve as strong retrievers for demonstrations even without fine-tuning on downstream tasks. Additionally, the pretrained weights can be utilized for fine-tuning the demonstration retriever, significantly reducing the amount of training data required compared to the pre-training phase, as we will show in §5.
> 9.	We added a sentence at the end of this paragraph: these works have shown that the diversity of demonstrations is crucial for achieving better in-context learning (ICL) performance.

---

> > ### Author Response · Authors · 2024-07-05
> > **Response to Reviewer m6ZC (Other points)**
> >
> > In the following, we address the comments on "other points"
> > 1.	The purpose of Table 1 is to provide a summary of each important paper and a guideline for readers to look at specific topics like the retriever training strategies and types. As you already notice that each column is related to the topic of the section outlined in the paper. In the beginning, we also considered splitting the table into each section, however, given the large amount of papers, this will take lots of space to have tables in each section.
> > 2.	Thank you for the suggestion, after careful consideration, we agree that One-hoc might not be an ideal phrase, and we change it to “Top-k” since this is the name used in standard IR systems and the process is the same as we mean in this paragraph. We have also added the citations in this paragraph to be consistent with other paragraphs in this section.
> > 3.	Naik et al work shows that the diversity of thoughts are important to achieve better performance for reasoning tasks. Since this work is built on top of CoT, therefore, we move this citation to the paragraph under “CoT” as follows: Naik et al. (2023) found that the diversity of reasoning path is important as well, therefore they propose DiversePrompting where an LLM is prompted to generate diverse reasoning path to solve a problem.
> > 4.	We have changed according to your suggestion, please see the format in our revised manuscript.

---

> ### Comment · Reviewer_m6ZC · 2024-07-16
> **Response to points 1-10**
>
> Thank you for your detailed read and clarifications and changes. I really do think they aid the reader for such a survey that will serve as a pointer to those new to the area. I'm happy with almost all the changes except the lingering ones below.
>
> 6. Very minor, but few-shot prompting is explicit instruction, I would rather highlight that the same methodology applies broadly.

---

> ### Comment · Reviewer_m6ZC · 2024-07-16
> **Response to Missing Analysis points**
>
> - 2: I still find this a bit unsatisfying, but it is now more clear.
> - 3: Thanks for the clarification, I think the addition is helpful.
> - 4: Minor, but "pusedo" -> "pseudo".
> - 5: I don't think this makes any more specific claims/suggestions/recommendations than the previous version, and some speculation here would be welcome (or pointers to active lines of research). How would one design a retriever around instructional content? What specifically is different?
> - 7: Perhaps it might be worth being more explicit: RetICL improves "on top of" CoT?

---

> > ### Author Response · Authors · 2024-07-23
> > **Follow-up to Reviewer's Response to Missing Analysis points**
> >
> > 4. We have fixed the typo.
> > 5. We have further made change to this paragraph, specifically: Demonstration Retriever Design Current demonstration retriever architectures are not significantly different from those used for raw text, with most efforts focusing on constructing the downstream training data. An open question remains: “How can retrievers be designed specifically for demonstrations rather than raw text?” A potential direction could involve training a task-aware retriever with instructions Asai et al. (2023), such that the fine-tuned retriever learns to tailor itself to instructional content. The instruction retriever should not only understand the semantic instructions but also retrieve the most relevant demonstrations from the database that follow these instructions.
> > 7: We have made further change in the revised paper, specifically: Such tasks are usually accompanied by CoT. Luo et al. (2023) demonstrate that retrieved demonstrations can be combined with the CoT technique to further enhance performance in mathematical reasoning tasks, showing that RetICL improves on top of CoT.

---

> ### Comment · Reviewer_m6ZC · 2024-07-16
> **Response to Other points**
>
> - 1: I'm still torn on the importance of this table, but I am not that tied to it. Perhaps it would be useful for a very familiar reader to use as a reference.
> - 3: This is clearer now, thanks.
> - 4: I would have kept the spacing of the previous version, and also kept both examples of the prompts for redundancy, but I would have simply bolded "[Task Instructions]" and numbered the versions as they are now. In the new revision I almost missed the "[Task Instructions]" prompt (which is just a reference to the previous version).

---

> > ### Author Response · Authors · 2024-07-23
> > **Follow-up to the reviewer's response to Other points.**
> >
> > 4: Thanks for the suggestion. We brought back both versions of the prompt. Following your suggestion, we also named the prompts so that in the third prompt we can directly refer to the second prompt. We also made the prompts look nicer visually. We are happy to update further if anything is still unclear. please refer to our new revision.

---

> > > ### Comment · Reviewer_m6ZC · 2024-07-24
> > > **Response to all lingering issues**
> > >
> > > Thanks to the authors for taking the comments so seriously. I do believe the paper has improved a lot from the initial submission and I'm happy to recommend acceptance.

---

> > > > ### Author Response · Authors · 2024-08-05
> > > >
> > > > Thanks for your recommendations and we appreciate your time and effort for reviewing our work.

---

> ### Author Response · Authors · 2024-07-23
> **Follow-up to reviewer's response to points 1-10**
>
> Thank you for your suggestion: We have further make the change in the revised version, specifically, Few-shot learning is a remarkable capability of LLMs, showcasing their generalization ability to rapidly adapt to a wide range of tasks with explicit instruction.

---

### Review · Reviewer_sZKR · 2024-06-21

**Summary Of Contributions:**

The paper presents a survey of in-context learning with retrieved demonstration. This field is concerned with performing in-context learning, in which, for a given query, a tailored set of examples is loaded.

The paper conceptualizes in-context learning (Sec 2), describing what drives the performance (Sec 2.2). Further, it introduces the setting of in-context learning with retrieved demonstration (RetICL) in Sec 3, describing its basic blocks: the retrieval objectives and strategies (Sec 3.1/Sec 3.2) and possible settings of retrieval corpus (Sec 3.3). The most important part of retrievers to be used with the discussion of pros and cons is contained in Section 4 and Section 5. In the subsequent Section 6, the paper discusses applications separately for NLP, reasoning, knowledge-based QA, and text generation. Finally, it is concluded with a discussion of open questions and future research directions.

**Audience:**

Yes

**Broader Impact Concerns:**

non

**Claims And Evidence:**

Yes

**Requested Changes:**

1. Perhaps it is worth explaining the relation of RetICL to RAG.
2. It is highly subjective, but I'd consider adding even more questions, perhaps even speculations, in the Section 7

**Strengths And Weaknesses:**

I found the paper well-structured and well-written. Basically, I do not see any major weaknesses.

---

> ### Author Response · Authors · 2024-07-05
> **Response to Reviewer sZKR**
>
> We really appreciate your high recommendation on our manuscript. The following is our response to your suggestions.
>
> Perhaps it is worth explaining the relation of RetICL to RAG.
> - Thanks for the suggestion, we added the following in the end of the last second paragraph in introduction: RetICL shares similarities with the broader concept of Retrieval Augmented Generation (RAG)~\citep{guu2020retrieval,lewis2020retrieval,izacard2023atlas}. Both approaches aim to retrieve external information to augment the prompt and improve model inference performance. However, RAG encompasses a wider spectrum of retrieval methods and use cases. While RetICL specifically focuses on retrieving demonstrations for in-context learning, RAG can involve retrieving various types of relevant information in response to a query, not necessarily limited to demonstrations.
>
> It is highly subjective, but I'd consider adding even more questions, perhaps even speculations, in the Section 7
> - We have changed section 7 with more specific suggestions and speculation, also, we added a new direction (Many-Shot Prompting).

---

### Decision · Action_Editor_vsSo · 2024-08-10

**Recommendation:** Accept as is

**Comment:**

The reviewers are in agreement that the paper is comprehensive, cleanly organized, and well-written. I therefore recommend survey certification for this work.

**Audience:**

In-context learning has received a great deal of recent interest and thus this survey, although focused on one specific aspect of ICL, is likely to have a sufficient audience.

**Claims And Evidence:**

This paper provides a survey of retrieval-based in-context learning (RetICL) for large language models. It summarizes the concept of RetICL and provides a detailed categorization of existing methods. Applications of RetICL, guidelines for choosing a method, and potential future directions are also discussed.